# CONTINUOUS GRAPH FLOW

## ABSTRACT

In this paper, we propose *Continuous Graph Flow*, a generative continuous flow based method that aims to model complex distributions of graph-structured data. Once learned, the model can be applied to an arbitrary graph, defining a probability density over the random variables represented by the graph. It is formulated as an ordinary differential equation system with shared and reusable functions that operate over the graphs. This leads to a new type of neural graph message passing scheme that performs *continuous message passing* over time. This class of models offers several advantages: a flexible representation that can generalize to variable data dimensions; ability to model dependencies in complex data distributions; reversible and memory-efficient; and exact and efficient computation of the likelihood of the data. We demonstrate the effectiveness of our model on a diverse set of generation tasks across different domains: graph generation, image puzzle generation, and layout generation from scene graphs. Our proposed model achieves significantly better performance compared to state-of-the-art models.

## 1 INTRODUCTION

Modeling and generating graph-structured data has important applications in various scientific fields such as building knowledge graphs (Lin et al., 2015; Bordes et al., 2011), inventing new molecular structures (Gilmer et al., 2017) and generating diverse images from scene graphs (Johnson et al., 2018). Being able to train expressive graph generative models is an integral part of AI research.

Significant research effort has been devoted in this direction. Traditional graph generative methods (Erdős & Rényi, 1959; Leskovec et al., 2010; Albert & Barabási, 2002; Airoldi et al., 2008) are based on rigid structural assumptions and lack the capability to learn from observed data. Modern deep learning frameworks within the variational autoencoder (VAE) (Kingma & Welling, 2014) formalism offer promise of learning distributions from data. Specifially, for structured data, research efforts have focused on bestowing VAE based generative models with the ability to learn structured latent space models (Lin et al., 2018; He et al., 2018; Kipf & Welling, 2016). Nevertheless, their capacity is still limited mainly because of the assumptions placed on the form of distributions. Another class of graph generative models are based on autoregressive methods (You et al., 2018; Kipf et al., 2018). These models construct graph nodes sequentially wherein each iteration involves generation of edges connecting a generated node in that iteration with the previously generated set of nodes. Such autoregressive models have been proven to be the most successful so far. However, due to the sequential nature of the generation process, the generation suffers from the inability to maintain long-term dependencies in larger graphs. Therefore, existing methods for graph generation are yet to realize the full potential of their generative power, particularly, the ability to model complex distributions with the flexibility to address variable data dimensions.

Alternatively, for modeling the relational structure in data, graph neural networks (GNNs) or message passing neural networks (MPNNs) (Scarselli et al., 2009; Gilmer et al., 2017; Duvenaud et al., 2015; Li et al., 2017; Kipf & Welling, 2017; Santoro et al., 2017; Zhang et al., 2018) have been shown to be effective in learning generalizable representations over variable input data dimensions. These models operate on the underlying principle of iterative *neural message passing* wherein the node representations are updated iteratively for a fixed number of steps. Hereafter, we use the term message passing to refer to this neural message passing in GNNs. We leverage this representational ability towards graph generation.

In this paper, we introduce a new class of models – *Continuous Graph Flow* (CGF): a graph generative model based on continuous normalizing flows (Chen et al., 2018; Grathwohl et al., 2019) that

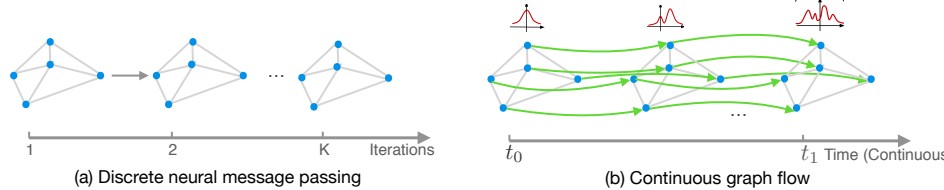

(a) Discrete neural message passing      (b) Continuous graph flow

Figure 1: Illustration of evolution of message passing mechanisms from discrete updates (a) to our proposed continuous updates (b). Continuous Graph Flow leverages normalizing flows to transform simple distributions (e.g. Gaussian) at $t_0$ to the target distributions at $t_1$. The distribution of only one graph node is shown here for visualization, but, all the node distributions transform over time.

generalizes the message passing mechanism in GNNs to continuous time. Specifically, to model continuous time dynamics of the graph variables, we adopt a neural ordinary different equation (ODE) formulation. Our CGF model has both the flexibility to handle variable data dimensions (by using GNNs) and the ability to model arbitrarily complex data distributions due to free-form model architectures enabled by the neural ODE formulation. Inherently, the ODE formulation also imbues the model with following properties: reversibility and exact likelihood computation.

Concurrent work on Graph Normalizing Flows (GNF) (Liu et al., 2019) also proposes a reversible graph neural network using normalizing flows. However, their model requires a fixed number of transformations. In contrast, while our proposed CGF is also reversible and memory efficient, the underlying flow model relies on continuous message passing scheme. Moreover, the message passing in GNF involves partitioning of data dimensions into two halves and employs coupling layers to couple them back. This leads to several constraints on function forms and model architectures that have a significant impact on performance (Kingma & Dhariwal, 2018). In contrast, our CGF model has unconstrained (free-form) Jacobians, enabling it to learn more expressive transformations. Moreover, other similar work GraphNVP Madhawa et al. (2019) is also based on normalizing flows as compared to CGF that models continuous time dynamics.

We demonstrate the effectiveness of our CGF-based models on three diverse tasks: graph generation, image puzzle generation, and layout generation based on scene graphs. Experimental results show that our proposed model achieves significantly better performance than state-of-the-art models.

## 2 PRELIMINARIES

**Graph neural networks.** Relational networks such as Graph Neural Networks (GNNs) facilitate learning of non-linear interactions using neural networks. In every layer $\ell \in \mathbb{N}$ of a GNN, the embedding $\boldsymbol{h}_i^{(\ell)}$ corresponding to a graph node accumulates information from its neighbors of the previous layer recursively as described below.

$$\boldsymbol{h}_i^{(\ell)} = g\left(\left\{f_{ij}(\boldsymbol{h}_i^{(\ell-1)}, \boldsymbol{x}_i^{(\ell-1)}, \boldsymbol{x}_j^{(\ell-1)})|j \in \mathcal{S}(i)\right\}\right), \tag{1}$$

where the function $g$ is an aggregator function, $\mathcal{S}(i)$ is the set of neighbour nodes of node $i$, and $f_{ij}$ is the message function from node $j$ to node $i$, $\boldsymbol{x}_i^{(\ell-1)}$ and $\boldsymbol{x}_j^{(\ell-1)}$ represent the node features corresponding to node $i$ and $j$ at layer $(\ell-1)$ respectively. Our model uses a restricted form of GNNs where embeddings of the graph nodes are updated in-place ($\boldsymbol{x}_i \leftarrow \boldsymbol{h}_i$), thus, we denote graph node as $\boldsymbol{x}$ and ignore $\boldsymbol{h}$ hereafter. These in-place updates allow using $\boldsymbol{x}_i$ in the flow-based models while maintaining the same dimensionality across subsequent transformations.

**Normalizing flows and change of variables.** Flow-based models enable construction of complex distributions from simple distributions (e.g. Gaussian) through a sequence of invertible mappings (Rezende & Mohamed, 2015). For instance, a random variable $\boldsymbol{z}$ is transformed from an initial state density $\boldsymbol{z}_0$ to the final state $\boldsymbol{z}_K$ using a chain of $K$ invertible functions $f_k$ described as:

$$\boldsymbol{z}_K = f_K \circ \dots f_2 \circ f_1(\boldsymbol{z}_0). \tag{2}$$

The computation of log-likelihood of a random variable uses change of variables rule formulated as:

$$\log p_K(\boldsymbol{z}_K) = \log p_0(\boldsymbol{z}_0) - \sum_{k=1}^{K} \log \left| \det \frac{\partial f_k(\boldsymbol{z}_{k-1})}{\partial \boldsymbol{z}_k} \right|, \tag{3}$$

where $\partial f_k(\boldsymbol{z}_{k-1})/\partial \boldsymbol{z}_k$ is the Jacobian of $f_k$ for $k \in \{1, 2, ..., K\}$.

**Continuous normalizing flows.** Continuous normalizing flows (CNFs) (Chen et al., 2018; Grathwohl et al., 2019) model the continuous-time dynamics by pushing the limit on number of transformations. Given a random variable $\boldsymbol{z}$, the following ordinary differential equation (ODE) defines the change in the state of the variable.

$$\frac{\partial \boldsymbol{z}}{\partial t} = f(z(t), t) \tag{4}$$

Chen et al. (2018) extended the change of variables rule described in Eq. 3 to continuous version. The dynamics of the log-likelihood of a random variable is then defined as the following ODE.

$$\frac{\partial p(\boldsymbol{z}(t))}{\partial t} = Tr\left(\frac{\partial f}{\partial \boldsymbol{z}(t)}\right) \tag{5}$$

Following the above equation, the log likelihood of the variable $z$ at time $t_1$ starting from time $t_0$ is

$$\log p(\boldsymbol{z}(t_1)) = \log p(\boldsymbol{z}(t_0)) - \int_{t_0}^{t_1} Tr\left(\frac{\partial f(\boldsymbol{z})}{\partial \boldsymbol{z}(t)}\right), \tag{6}$$

where the trace computation is more computationally efficient than computation of the Jacobian in Equation (4). Building on CNFs, we present continuous graph flow which effectively models continuous time dynamics over graph-structured data.

## 3 CONTINUOUS GRAPH FLOW

Given a set of random variables $\boldsymbol{X}$ containing $n$ related variables, the goal is to learn the joint distribution $p(\boldsymbol{X})$ of the set of variables $\boldsymbol{X}$. Each element of set $\boldsymbol{X}$ is $\boldsymbol{x}_i \in \mathbb{R}^m$ where $i = 1, 2 \ldots, n$ and $m$ represents the number of dimensions of the variable. For continuous time dynamics of the set of variables $\boldsymbol{X}$, we formulate an ordinary differential equation system as follows:

$$\begin{bmatrix} \dot{\boldsymbol{x}}_1(t) \\ \dot{\boldsymbol{x}}_2(t) \\ \vdots \\ \dot{\boldsymbol{x}}_n(t) \end{bmatrix} = \begin{bmatrix} f^1(\boldsymbol{X}(t)) \\ f^2(\boldsymbol{X}(t)) \\ \vdots \\ f^n(\boldsymbol{X}(t)) \end{bmatrix}, \tag{7}$$

where $\dot{\boldsymbol{x}_i} = d\boldsymbol{x}_i/dt$ and $\boldsymbol{X}(t)$ is the set of variables at time $t$. The random variable $\boldsymbol{x}_i$ at time $t_0$ follows a base distribution that can have simple forms, *e.g.* Gaussian distributions. The function $f^i$ implicitly defines the interaction among the variables. Following this formulation, the transformation of the individual graph variable is defined as

$$\boldsymbol{x}_i(t_1) = \boldsymbol{x}_i(t_0) + \int_{t_0}^{t_1} f^i(\boldsymbol{X}(t))dt, \tag{8}$$

This provides transformation of the value of the variable $\boldsymbol{x}_i$ from time $t_0$ to time $t_1$.

### 3.1 CONTINUOUS MESSAGE PASSING

The form in Eq. 8 represents a generic multi-variate update where interaction functions are defined over all the variables in the set $\boldsymbol{X}$. However, the functions do not take into account the relational structure between the graph variables.

To address this, we define a neural message passing process that operates over a graph by defining the update functions over variables according to the graph structure. This process begins from time $t_0$ where each variable $\boldsymbol{x}_i(t_0)$ only contain local information. At time $t$, these variables are updated based on the information gathered from other neighboring variables. For such updates, the function $f^i$ in Eq. 8 is defined as:

$$f^i(\boldsymbol{X}(t)) = g(\{\hat{f}_{ij}(\boldsymbol{x}_i(t), \boldsymbol{x}_j(t))|j \in \mathcal{S}(i)\}), \tag{9}$$

where $\hat{f}_{ij}(\cdot)$ is a reusable message function and used for passing information between variables $\boldsymbol{x_i}$ and $\boldsymbol{x_j}$, $\mathcal{S}(i)$ is the set of neighboring variables that interacts with variable $\boldsymbol{x_i}$, and $g(\cdot)$ is a function

that aggregates the information passed to a variable. The above formulation describes the case of pairwise message functions, though this can be generalized to higher-order interactions.

We formulate it as a continuous process which eliminates the requirement of having a predetermined number of steps of message passing. By further pushing the message passing process to update at infinitesimally smaller steps and continuing the updates for an arbitrarily large number of steps, the update associated with each variable can be represented using shared and reusable functions as the following ordinary differential equation (ODE) system.

$$
\begin{bmatrix} \dot{\boldsymbol{x}}_1(t) \\ \dot{\boldsymbol{x}}_2(t) \\ \vdots \\ \dot{\boldsymbol{x}}_n(t) \end{bmatrix} = \begin{bmatrix} g(\{\hat{f}_{1j}(\boldsymbol{x}_1(t), \boldsymbol{x}_j(t)) | j \in \mathcal{S}(1)\}) \\ g(\{\hat{f}_{2j}(\boldsymbol{x}_2(t), \boldsymbol{x}_j(t)) | j \in \mathcal{S}(2)\}) \\ \vdots \\ g(\{\hat{f}_{nj}(\boldsymbol{x}_n(t), \boldsymbol{x}_j(t)) | j \in \mathcal{S}(n)\}) \end{bmatrix},
\tag{10}
$$

where $\dot{\boldsymbol{x}_i} = d\boldsymbol{x}_i/dt$. Performing message passing to derive final states is equivalent to solving an initial value problem for an ODE system. Following the ODE formulation, the final states of variables can be computed as follows. This formulation can be solved with an ODE solver.

$$
\boldsymbol{x}_i(t_1) = \boldsymbol{x}_i(t_0) + \int_{t_0}^{t_1} g\left(\left\{\hat{f}_{ij}(\boldsymbol{x}_i(t), \boldsymbol{x}_j(t)) | j \in \mathcal{S}(i)\right\}\right), i \in \{1, \ldots, n\}.
\tag{11}
$$

### 3.2 Continuous message passing for density transformation

Continuous graph flow leverages the continuous message passing mechanism (described in Sec. 3.1) and formulates the message passing as implicit density transformations of the variables (illustrated in Figure 1). Given a set of variables $\boldsymbol{X}$ with dependencies among them, the goal is to learn a model that captures the distribution from which the data were sampled. Assume the joint distribution $p(\boldsymbol{X})$ at time $t_0$ has a simple form such as independent Gaussian distribution for each variable $\boldsymbol{x}_i(t_0)$. The continuous message passing process allows the transformation of the set of variables from $\boldsymbol{X}(t_0)$ to $\boldsymbol{X}(t_1)$. Moreover, this process also converts the distributions over variables from simple base distributions to complex data distributions. Building on the *independent variable* continuous time dynamics described in Eq. 5, we define the dynamics corresponding to related *graph variables* as:

$$
\frac{\partial \log p(\boldsymbol{X}(t))}{\partial t} = -Tr\left(\frac{\partial F}{\partial \boldsymbol{X}(t)}\right),
\tag{12}
$$

where $F$ represents a set of reusable functions incorporating aggregated messages. Therefore, the joint distribution of set of variables $\boldsymbol{X}$ can be described as:

$$
\log p(\boldsymbol{X}(t_1)) = \log p(\boldsymbol{X}(t_0)) - \int_{t_0}^{t_1} Tr\left(\frac{\partial F}{\partial \boldsymbol{X}(t)}\right).
\tag{13}
$$

Here we use two types of density transformations for message passing: (1) *generic message transformations* – transformations with generic update functions where trace in Eq. 13 can be approximated instead of computing by brute force method, and (2) *multi-scale message transformations* – transformations with generic update functions at multiple scales of information.

**Generic message transformations.** The trace of Jacobian matrix in Eq. 13 is modeled using a generic neural network function. The likelihood is defined as:

$$
\log p(\boldsymbol{X}(t_1)) = \log p(\boldsymbol{X}(t_0)) - \mathbb{E}_{p(\boldsymbol{\epsilon})} \int_{t_0}^{t_1} \left[\boldsymbol{\epsilon}^T \frac{\partial F}{\partial \boldsymbol{X}(t)} \boldsymbol{\epsilon} dt\right],
\tag{14}
$$

where $F$ denotes a neural network for message functions, and $\boldsymbol{\epsilon}$ is a noise vector and usually can be sampled from standard Gaussian or Rademacher distributions.

**Multi-scale message transformations.** As a generalization of generic message transformations, we design a model with multi-scale message passing to encode different levels of information in the variables. Similar to Dinh et al. (2016), we construct our multi-scale CGF model by stacking

several blocks wherein each flow block performs message passing based on generic message transformations. After passing the input through a block, we factor out a portion of the output and feed it as input to the subsequent block. The likelihood is defined as:

$$\log p(\boldsymbol{X}(t_b)) = \log p(\boldsymbol{X}(t_{b-1})) - \mathbb{E}_{p(\boldsymbol{\epsilon})} \int_{t_{b-1}}^{t_b} \left[ \boldsymbol{\epsilon}^T \frac{\partial F}{\partial \boldsymbol{X}(t_{b-1})} \boldsymbol{\epsilon} dt \right], \tag{15}$$

where $b = 1, 2, \ldots, (B - 1)$ with $B$ as the total number of blocks in the design of the multi-scale architecture. Assume at time $t_b$ $(t_0 < t_b < t_1)$, $\boldsymbol{X}(t_b)$ is factored out into two. We use one of these (denoted as $\tilde{\boldsymbol{X}}(t_b)$) as the input to the $(b + 1)^{\text{th}}$ block. Let $\tilde{\boldsymbol{X}}(t_b)$ be the input to the next block, the density transformation is formulated as:

$$\log p(\tilde{\boldsymbol{X}}(t_{b+1})) = \log p(\tilde{\boldsymbol{X}}(t_b)) - \mathbb{E}_{p(\boldsymbol{\epsilon})} \int_{t_b}^{t_{b+1}} \left[ \boldsymbol{\epsilon}^T \frac{\partial F}{\partial \tilde{\boldsymbol{X}}(t_b)} \boldsymbol{\epsilon} dt \right]. \tag{16}$$

## 4 EXPERIMENTS

To demonstrate the generality and effectiveness of our Continuous Graph Flow (CGF), we evaluate our model on three diverse tasks: (1) graph generation, (2) image puzzle generation, and (3) layout generation based on scene graphs. Graph generation requires the model to learn complex distributions over the graph structure. Image puzzle generation requires the model to learn local and global correlations in the puzzle pieces. Layout generation has a diverse set of possible nodes and edges. These tasks have high complexity in the distributions of graph variables and diverse potential function types. Together these tasks pose a challenging evaluation for our proposed method.

### 4.1 GRAPH GENERATION

**Datasets and Baselines.** We evaluate our model on graph generation on two benchmark datasets EGO-SMALL and COMMUNITY-SMALL (You et al., 2018) against four strong state-of-the-art baselines: VAE-based method (Simonovsky & Komodakis, 2018), autoregressive graph generative model GraphRNN (You et al., 2018) and DeepGMG (Li et al., 2018), and Graph normalizing flows (Liu et al., 2019).

**Evaluation.** We conduct a quantitative evaluation of the generated graphs using Maximum Mean Discrepancy (MMD) measures proposed in GRAPHRNN (You et al., 2018). The MMD evaluation in GRAPHRNN was performed using a test set of N ground truth graphs, computing their distribution over the nodes, and then searching for a set of N generated graphs from a larger set of samples generated from the model that best matches this distribution. As mentioned by Liu et al. (2019), this evaluation process would likely have high variance as the graphs are very small. Therefore, we also performed an evaluation by generating 1024 graphs for each model and computing the MMD distance between this generated set of graphs and the ground truth test set. Baseline results are from Liu et al. (2019). Implementation details refer to Appendix A.

**Results and Analysis.** Table 1 shows the results in terms of MMD. Our CGF outperforms the baselines by a significant margin and also the concurrent work GNF. We believe our CGF outperforms GNF because it employs free-flow functions forms unlike GNF that has some contraints necessitated by the coupling layers. Fig. 2 visualizes the graphs generated by our model. Our model can capture the characteristics of datasets and generate diverse graphs that are not seen during the training. For additional visualizations and comparisons, refer to the Appendix A.

### 4.2 IMAGE PUZZLE GENERATION

**Task description.** We design image puzzles for image datasets to test model's ability on fitting very complex node distributions in graphs. Given an image of size $W \times W$, we design a puzzle by dividing the original image into non-overlapping unique patches. A puzzle patch is of size $w \times w$, in which $w$ represents the width of the puzzle. Each image is divided into $p = W/w$ puzzle patches both horizontally and vertically, and therefore we obtain $P = p \times p$ patches in total. Each patch corresponds to a node in the graph. To evaluate the performance of our model on dynamic graph sizes, instead of training the model with all nodes, we sample $\tilde{p}$ adjacent patches where $\tilde{p}$ is uniformly sampled from $\{1, \ldots, P\}$ as input to the model during training and test. In our experiments, we use

Table 1: **Quantitative results on graph generation.** MMD measures between test set and generated graphs (lower is better). The second set shows GRAPHRNN evaluation with node distribution matching (averaged over 5 different models with 3 trials). The third set shows evaluation for the test set for all 1024 generated graphs (averaged over 5 models). $(\max(|V|), \max(|E|))$ is also shown.

| Method | COMMUNITY-SMALL (20,83) | | | EGO-SMALL (18,69) | | |
|---|---|---|---|---|---|---|
| | DEGREE | CLUSTERING | ORBIT | DEGREE | CLUSTERING | ORBIT |
| GRAPHVAE | 0.35 | 0.98 | 0.54 | 0.13 | 0.17 | 0.05 |
| DEEPGMG | 0.22 | 0.95 | 0.4 | 0.04 | 0.10 | 0.02 |
| GRAPHRNN | 0.08 | 0.12 | 0.04 | 0.09 | 0.22 | 0.003 |
| GNF + AE | 0.20 | 0.20 | 0.11 | 0.03 | 0.10 | 0.001 |
| **CGF** | 0.10 | 0.30 | 0.08 | 0.02 | 0.11 | 0.001 |
| GRAPHRNN (1024) | 0.03 | **0.01** | 0.01 | 0.04 | 0.05 | 0.06 |
| GNF + AE (1024) | 0.12 | 0.15 | 0.02 | 0.01 | 0.03 | 0.0008 |
| **CGF (1024)** | **0.02** | 0.02 | **0.001** | **0.002** | **0.007** | **0.0002** |

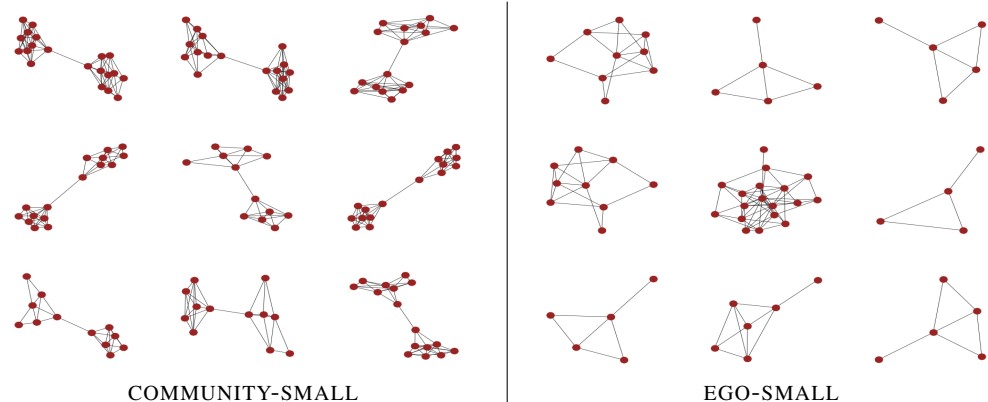

COMMUNITY-SMALL          EGO-SMALL

Figure 2: **Visualization of generated graphs from our model.** Our model can capture the characteristic of datasets and generate diverse graphs not appearing in the training set.

patch size $w = 16$, $p \in \{2, 3, 4\}$ and edge function for each direction (*left*, *right*, *up*, *down*) within a neighbourhood of a node. Additional details are in Appendix A.

**Datasets and baselines.** We design the image puzzle generation task for three datasets: MNIST (LeCun et al., 1998), CIFAR10 (Krizhevsky et al., 2009), and CelebA (Liu et al., 2015). CelebA dataset does not have a validation set, thus, we split the original dataset into a training set of 27,000 images and test set of 3,000 images as in (Kingma & Dhariwal, 2018). We compare our model with six state-of-the-art VAE based models: (1) StructuredVAE (He et al., 2018), (2) Graphite (Grover et al., 2019), (3) Variational message passing using structured inference networks (VMP-SIN) (Lin et al., 2018), (4) BiLSTM + VAE: a bidirectional LSTM used to model the interaction between node latent variables (obtained after serializing the graph) in an autoregressive manner similar to Gregor et al. (2015), (5) Variational graph autoencoder (GAE) (Kipf & Welling, 2016), and (6) Neural relational inference (NRI) (Kipf et al., 2018): we adapt this to model data for single time instance and model interactions between the nodes.

**Results and analysis.** We report the negative log likelihood (NLL) in bits/dimension (lower is better). The results in Table 2 indicate that CGF significantly outperforms the baselines. In addition to the quantitative results, we also conduct sampling based evaluation and perform two types of generation experiments: (1) *Unconditional Generation:* Given a puzzle size $p$, $p^2$ puzzle patches are generated using a vector $z$ sampled from Gaussian distribution (refer Fig. 3(a)); and (2) *Conditional Generation:* Given $p_1$ patches from an image puzzle having $p^2$ patches, we generate the remaining $(p^2 - p_1)$ patches of the puzzle using our model (see Fig. 3(b)). We believe the task of conditional generation is easier than unconditional generation as there is more relevant information in the input during flow based transformations. For unconditional generation, samples from a base distribution (e.g. Gaussian) are transformed into learnt data distribution using the CGF model. For conditional generation, we map $\boldsymbol{x}_a \in \boldsymbol{X}_a$ where $\boldsymbol{X}_a \subset \boldsymbol{X}$ to the points in base distribution to obtain $\boldsymbol{z}_a$ and

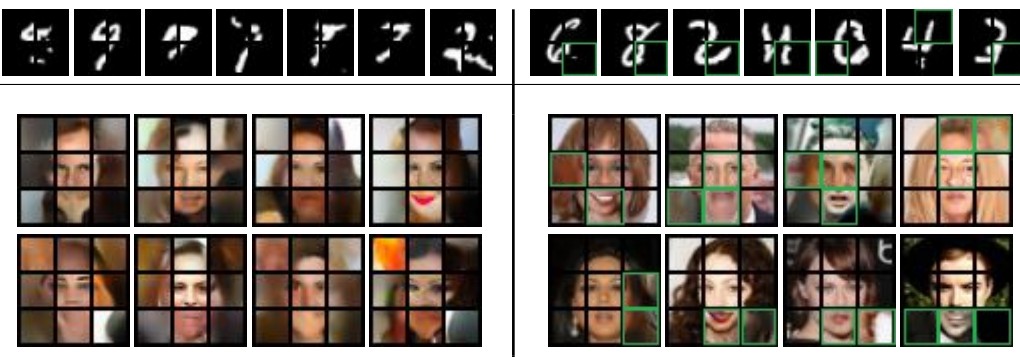

| (a) Unconditional Generation | (b) Conditional Generation |

Figure 3: **Qualitative results for image puzzle generation.** Samples generated using our model for 2x2 MNIST puzzles (above horizontal line) and 3x3 CelebA-HQ puzzles (below horizontal line) in (a) *unconditional generation* and (b) *conditional generation* settings. For setting (b), generated patches (highlighted in green boxes) are conditioned on the remaining patches (from ground truth).

subsequently concatenate the samples from Gaussian distribution to $z_a$ to obtain $z'$ that match the dimensions of desired graph and generate samples by transforming from $z'$ to $x \in X$ using the trained graph flow.

Table 2: **Quantitative results on image puzzle generation.** Comparison of our CGF model with standard VAE and state-of-the-art VAE based models in bits/dimension (lower is better). These results are for unconditional generation using multi-scale version of continuous graph flow.

| Method | MNIST | | | CIFAR-10 | | | CelebA-HQ | | |
|---|---|---|---|---|---|---|---|---|---|
| | 2x2 | 3x3 | 4x4 | 2x2 | 3x3 | 4x4 | 2x2 | 3x3 | 4x4 |
| BiLSTM + VAE | 4.97 | 4.77 | 4.42 | 6.02 | 5.20 | 4.53 | 5.72 | 5.66 | 5.48 |
| StructuredVAE (He et al., 2018) | 4.89 | 4.65 | 3.82 | 6.03 | 5.02 | 4.70 | 5.66 | 5.43 | 5.27 |
| Graphite (Grover et al., 2019) | 4.90 | 4.64 | 4.02 | 6.06 | 5.09 | 4.61 | 5.71 | 5.50 | 5.32 |
| VMP-SIN (Lin et al., 2018) | 5.13 | 4.92 | 4.44 | 6.00 | 4.96 | 4.34 | 5.70 | 5.43 | 5.27 |
| GAE (Kipf & Welling, 2016) | 4.91 | 4.89 | 4.17 | 5.83 | 4.95 | 4.21 | 5.71 | 5.63 | 5.28 |
| NRI (Kipf et al., 2018) | 4.58 | 4.35 | 4.11 | 5.44 | 4.82 | 4.70 | 5.36 | 5.43 | 5.28 |
| CGF | **1.24** | **1.21** | **1.20** | **2.42** | **2.31** | **2.00** | **3.44** | **3.17** | **3.16** |

## 4.3 LAYOUT GENERATION FROM SCENE GRAPHS

**Task description and evaluation metrics.** Layout generation from scene graphs is a crucial task in computer vision and bridges the gap between the symbolic graph-based scene description and the object layouts in the scene (Johnson et al., 2018; Zhao et al., 2019; Jyothi et al., 2019). Scene graphs represent scenes as directed graphs, where nodes are objects and edges give relationships between objects. Object layouts are described by the set of corresponding bounding box annotations (Johnson et al., 2018). Our model uses scene graph as inputs (nodes correspond to objects and edges represent relations). An edge function is defined for each relationship type. The output contains a set of object bounding boxes described by $\{[x_i, y_i, h_i, w_i]\}_{i=1}^n$, where $x_i, y_i$ are the top-left coordinates, and $w_i, h_i$ are the bounding box width and height respectively. We use negative log likelihood per node (lower is better) for evaluating models on scene layout generation.

**Datasets and baselines.** Two large-scale challenging datasets are used to evaluate the proposed model: Visual Genome (Krishna et al., 2017) and COCO-Stuff (Caesar et al., 2018) datasets. Visual Genome contains 175 object and 45 relation types. The training, validation and test set contain 62565, 5506 and 5088 images respectively. COCO-Stuff dataset contains 24972 train, 1024 validation, and 2048 test scene graphs. We use the same baselines as in Sec. 4.2.

**Results and analysis.** We show quantitative results in Table 3 against several state-of-the-art baselines. Our CGF model significantly outperforms these baselines in terms of negative log likelihood. Moreover, we show some qualitative results in Fig. 4. Our model can learn the correct relations defined in scene graphs for both conditional and unconditional generation, Furthermore, our model is capable to learn *one-to-many* mappings and generate diverse of layouts for the same scene graph.

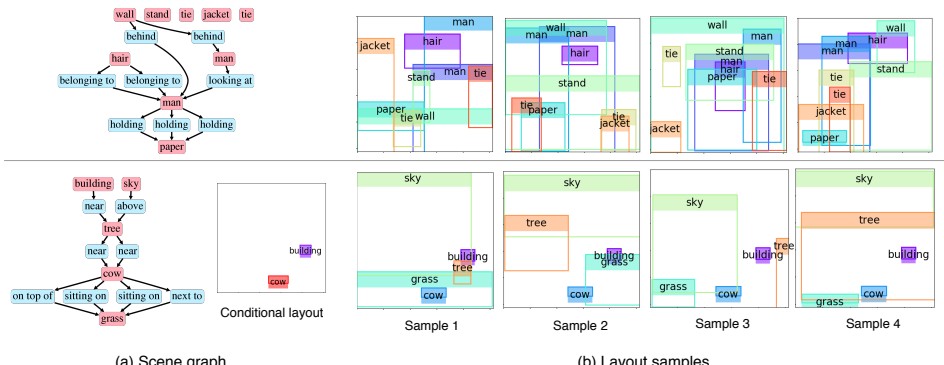

(a) Scene graph             (b) Layout samples

Figure 4: **Visualization for layout generation** on Visual Genome. Our CGF model can generate diverse layouts for the same scene graph. Upper row: layout samples with unconditional generation. Lower row: Layout generation conditioned on known layout. Best viewed in color.

Table 3: **Quantitative results for layout generation for scene graph** in negative log-likelihood. These results are for unconditional generation using CGF with generic message transformations.

| Method | Visual Genome | COCO-Stuff |
|---|---|---|
| BiLSTM + VAE | -1.20 | -1.60 |
| StructuredVAE (He et al., 2018) | -1.05 | -1.36 |
| Graphite (Grover et al., 2019) | -1.17 | -0.93 |
| VMP-SIN (Lin et al., 2018) | -0.61 | -0.85 |
| GAE (Kipf & Welling, 2016) | -1.85 | -1.92 |
| NRI (Kipf et al., 2018) | -0.76 | -0.91 |
| CGF | **-4.24** | **-6.21** |

## 4.4 ANALYSIS: GENERALIZATION TEST

To test the generalizability of our model to variable graph sizes, we design three different evaluation settings and test it on image puzzle task: (1) *odd to even*: training with graphs having odd graph sizes and testing on graphs with even numbers of nodes, (2) *less to more*: training on graphs with smaller sizes and testing on graphs with larger sizes, and (3) *more to less*: training on graphs with larger sizes and testing on graphs with smaller. In the *less to more* setting, we test the model's ability to use the functions learned from small graphs on more complicated ones, whereas the *more to less* setting evaluates the model's ability to learn disentangled functions without explicitly seeing them during training. In our experiments, for the *less to more* setting, we use sizes less than $G/2$ for training and more than $G/2$ for testing where G is the size of the full graph. Similarly, for the *less to more* setting, we use sizes less than $G/2$ for training and more than $G/2$ for testing. Table 4 reports the NLL for these settings. The NLL of these models are close to the performance on the models trained on full dataset demonstrating that our model is able to generalize to unseen graph sizes.

Table 4: **Generalization test** in three different evaluation settings for image puzzle sizes 3x3 for three image datasets in bits/dimension.

| Settings | MNIST | CIFAR-10 | CelebA-HQ |
|---|---|---|---|
| Odd to even | 1.33 | 2.81 | 3.31 |
| Less to more | 1.37 | 2.91 | 3.66 |
| More to less | 1.34 | 2.83 | 3.44 |

## 5 CONCLUSION

In this paper, we presented *continuous graph flow*, a generative model that generalizes the neural message passing in graphs to continuous time. We formulated the model as an neural ordinary differential equation system with shared and reusable functions that operate over the graph structure. We conducted evaluation for a diverse set of generation tasks across different domains: graph generation, image puzzle generation, and layout generation for scene graph. Experimental results showed that continuous graph flow achieves significant performance improvement over various of state-of-the-art baselines. For future work, we will focus on generation tasks for large-scale graphs which is promising as our model is reversible and memory-efficient.

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

# A   APPENDIX

We provide supplementary materials to support the contents of the main paper. In this part, we describe implementation details of our model. We also provide additional qualitative results for the generation tasks: Graph generation, image puzzle generation and layout generation from scene graphs.

## A.1   IMPLEMENTATION DETAILS

The ODE formulation for continuous graph flow (CGF) model was solved using ODE solver provided by NeuralODE (Chen et al., 2018). In this section, we provide specific details of the configuration of our CGF model used in our experiments on two different generation tasks used for evaluation in the paper.

**Graph Generation.** For each graph, we firstly generate its line graph with edges switched to nodes and nodes switched to edges. Then the graph generation problem is now generating the current nodes values which represents the adjacency matrix in the original graph. Each node value is binary (0 or 1) and is dequantized to continuous values through variational dequantization, with a global learnable Gaussian distribution as variational distribution. For our architecture, we use two blocks of continuous graph flow with two fully connected layers in Community-small dataset, and one block of continuous graph flow with one fully connected layer in Citeseer-small dataset. The hidden dimensions are all 32.

**Image puzzle generation.** Each graph for this task comprise nodes corresponding to the puzzle pieces. The pieces that share an edge in the puzzle grid are considered to be connected and an edge function is defined over those connections. In our experiments, each node is transformed to an embedding of size 64 using convolutional layer. The graph message passing is performed over these node embeddings. The image puzzle generation model is designed using a multi-scale continuous graph flow architecture. We use two levels of downscaling in our model each of which factors out the channel dimension of the random variable by 2. We have two blocks of continuous graph flow before each downscaling wth four convolutional message passing blocks in each of them. Each message passing block has a unary message passing function and binary passing functions based on the edge types – all containing hidden dimensions of 64.

**Layout generation for scene graphs.** For scene graph layout generation, a graph comprises node corresponding to object bounding boxes described by $\{[x_i, y_i, h_i, w_i]\}_{i=1}^n$, where $x_i, y_i$ represents the top-left coordinates, and $w_i, h_i$ represents the bounding box width and height respectively and edge functions are defined based on the relation types. In our experiments, the layout generation model uses two blocks of continuous graph flow units, with four linear graph message passing blocks in each of them. The message passing function uses 64 hidden dimensions, and takes the embedding of node label and edge label in unary message passing function and binary message passing function respectively. The embedding dimension is also set to 64 dimensions. For binary message passing function, we pass the messages both through the direction of edge and the reverse direction of edge to increase the model capacity.

## A.2   IMAGE PUZZLE GENERATION: ADDITIONAL QUALITATIVE RESULTS FOR CELEBA-HQ

Fig. 5 and Fig. 6 presents the image puzzles generated using unconditional generation and conditional generation respectively.

## A.3   LAYOUT GENERATION FROM SCENE GRAPH: QUALITATIVE RESULTS

Fig. 7 and Fig. 8 show qualitative result on unconditional and conditional layout generation from scene graphs for COCO-stuff dataset respectively. Fig. 9 and Fig. 10 show qualitative result on unconditional and conditional layout generation from scene graphs for Visual Genome dataset respectively. The generated results have diverse layouts corresponding to a single scene graph.

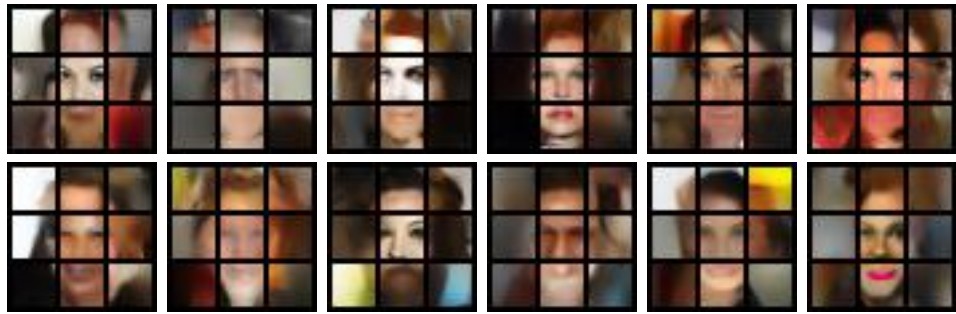

Figure 5: **Qualitative results on CelebA-HQ for image puzzle generation.** Samples generated using our model for 3x3 CelebA-HQ puzzles in unconditional generation setting. Best viewed in color.

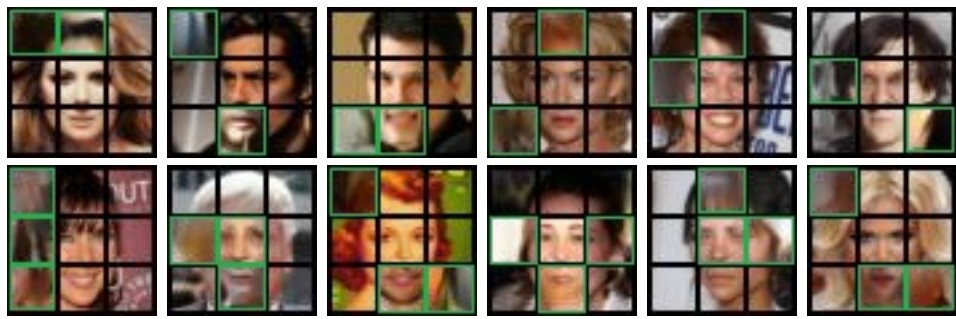

Figure 6: **Qualitative results on CelebA-HQ for image puzzle generation.** Samples generated using our model for 3x3 CelebA-HQ puzzles in conditional generation setting. Generated patches are highlighted in green. Best viewed in color.

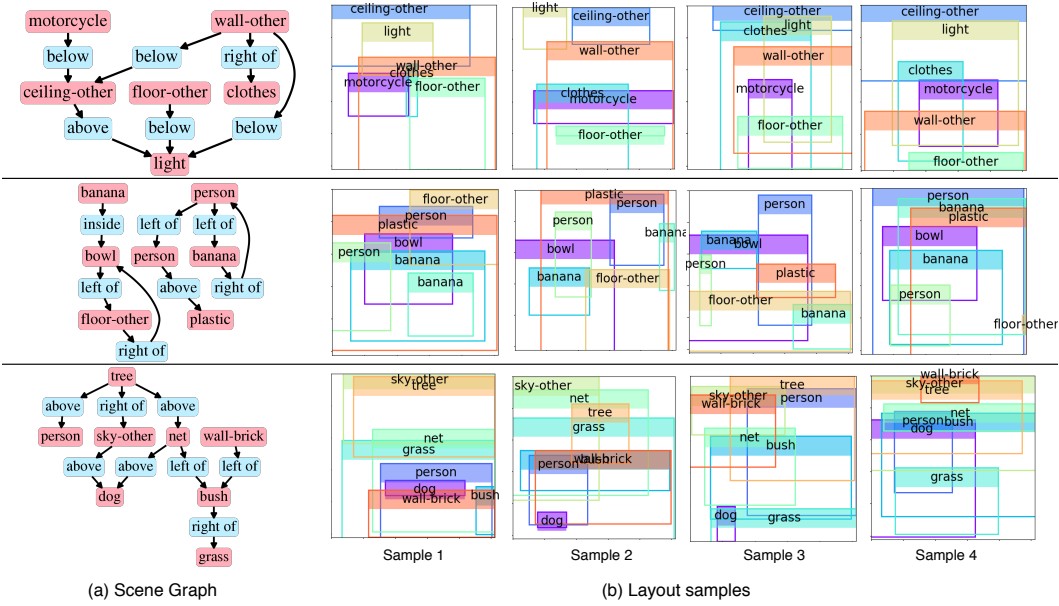

Figure 7: **Examples of *Unconditional* generation of layouts from scene graphs for COCO-Stuff dataset**. We sample 4 layouts. The generated results have different layouts, but sharing the same scene graph. Best viewed in color. Please zoom in to see the category of each object.

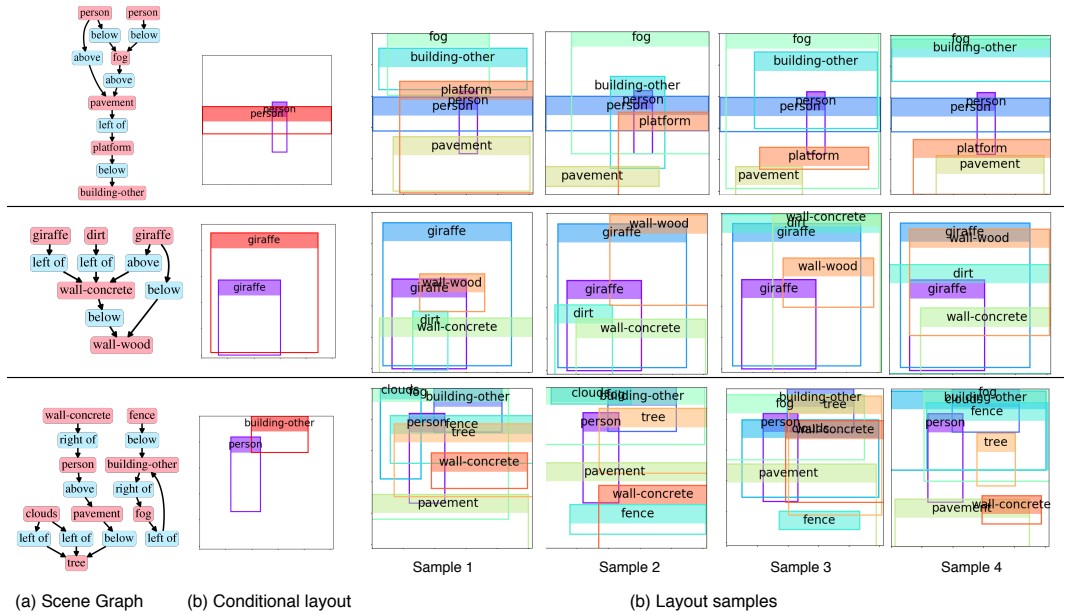

Figure 8: ***Conditional* generation of layouts from scene graphs for COCO-stuff dataset**. We sample 4 layouts. The generated results have different layouts except the conditional layout objects in (b), but sharing the same scene graph. Best viewed in color. Please zoom in to see the category of each object.

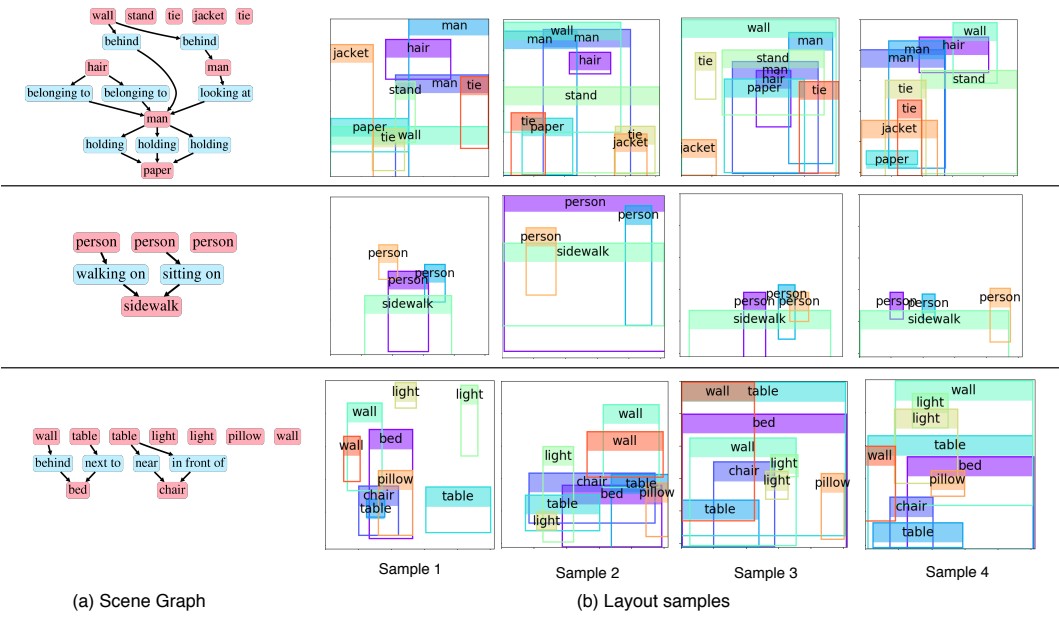

Figure 9: ***Unconditional* generation of layouts from scene graphs for Visual Genome dataset**. We sample 4 layouts for each scene graph. The generated results have different layouts, but sharing the same scene graph. Best viewed in color. Please zoom in to see the category of each object.

## A.4   GRAPH GENERATION: ADDITIONAL QUALITATIVE RESULTS

Fig. 11 and Fig. 12 present the generated graphs for EGO-SMALL and COMMUNITY-SMALL respectively.

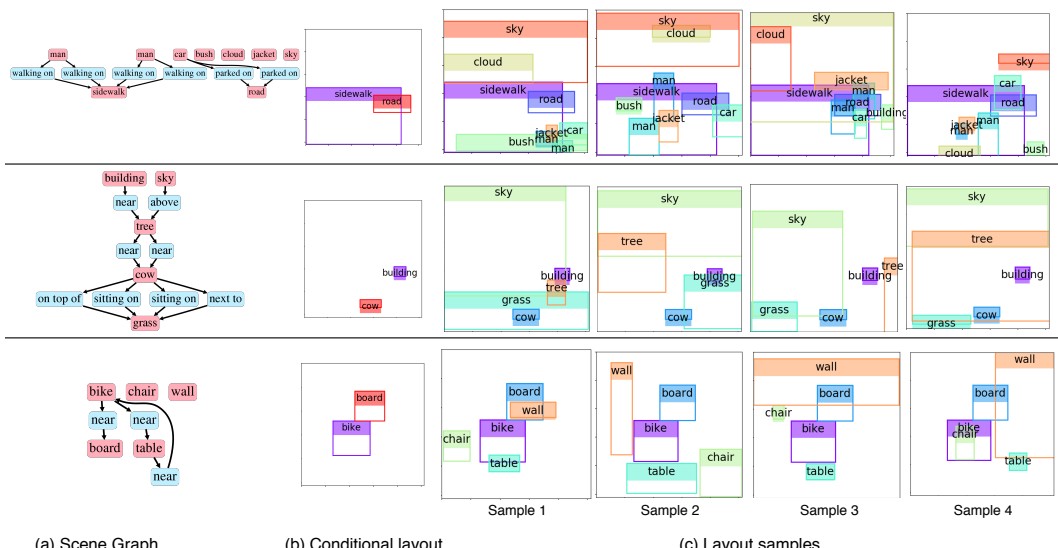

(a) Scene Graph      (b) Conditional layout      (c) Layout samples

Figure 10: *Conditional* **generation of layouts from scene graphs for Visual Genome dataset**. We sample 4 layouts for each scene graph. The generated results have different layouts except the conditional layout objects in (b), but sharing the same scene graph. Best viewed in color. Please zoom in to see the category of each object.

## A.5 ANALYSIS: NUMBER OF FUNCTION EVALUATION VS NUMBER OF NODES

We analyze the variation in number of function evaluations (NFE) required to solve the ODE as the number of nodes in the graph changes. Refer to Figure 13. The results interestingly show that the average number of function evaluation does not increase linearly with the increment in number of graph nodes, which would be the case if the variables were independent.

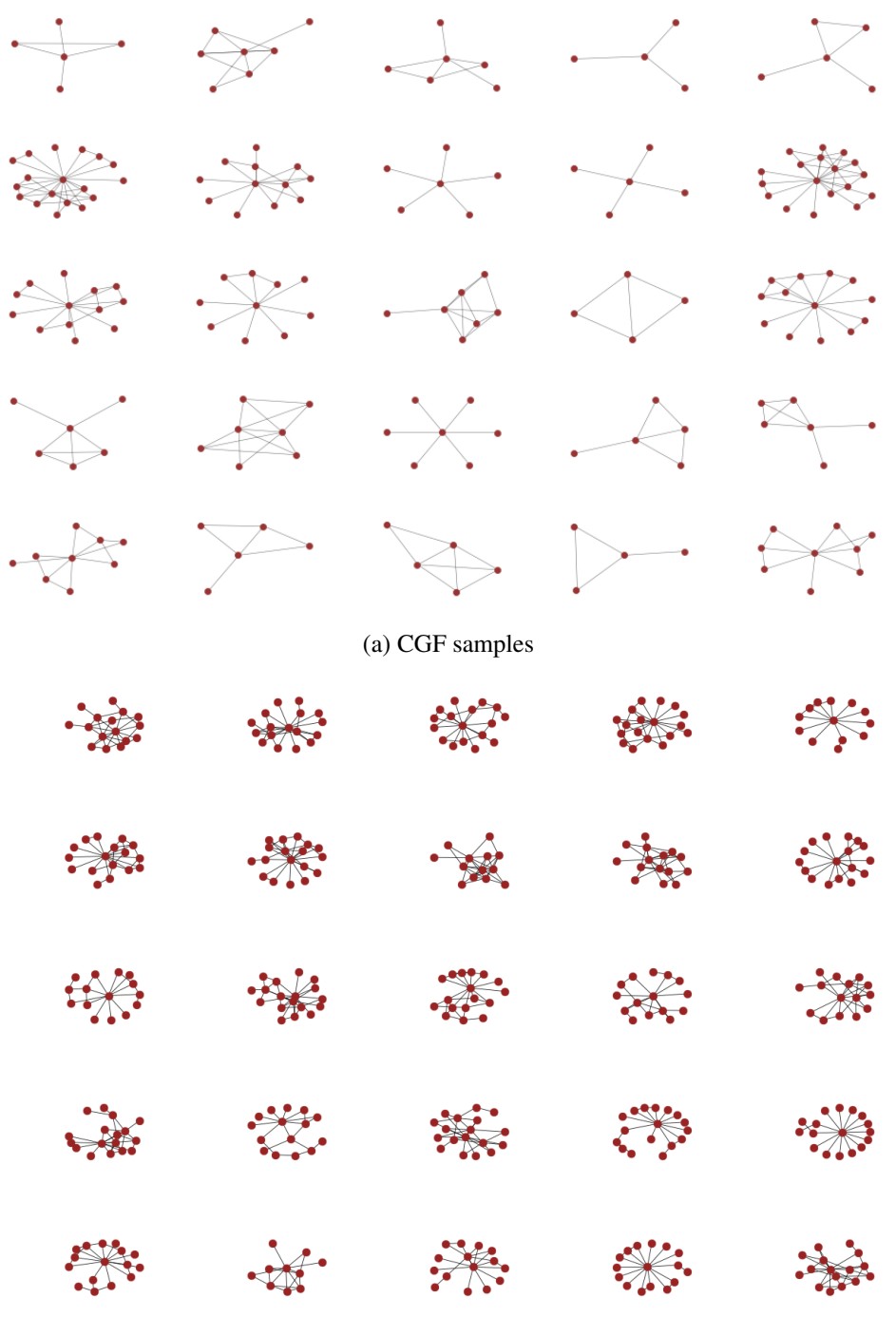

(a) CGF samples

(b) GRAPHRNN SAMPLES

Figure 11: **Graph generation on EGO-SMALL.** Samples generated using (a) our CGF and (b) GRAPHRNN.

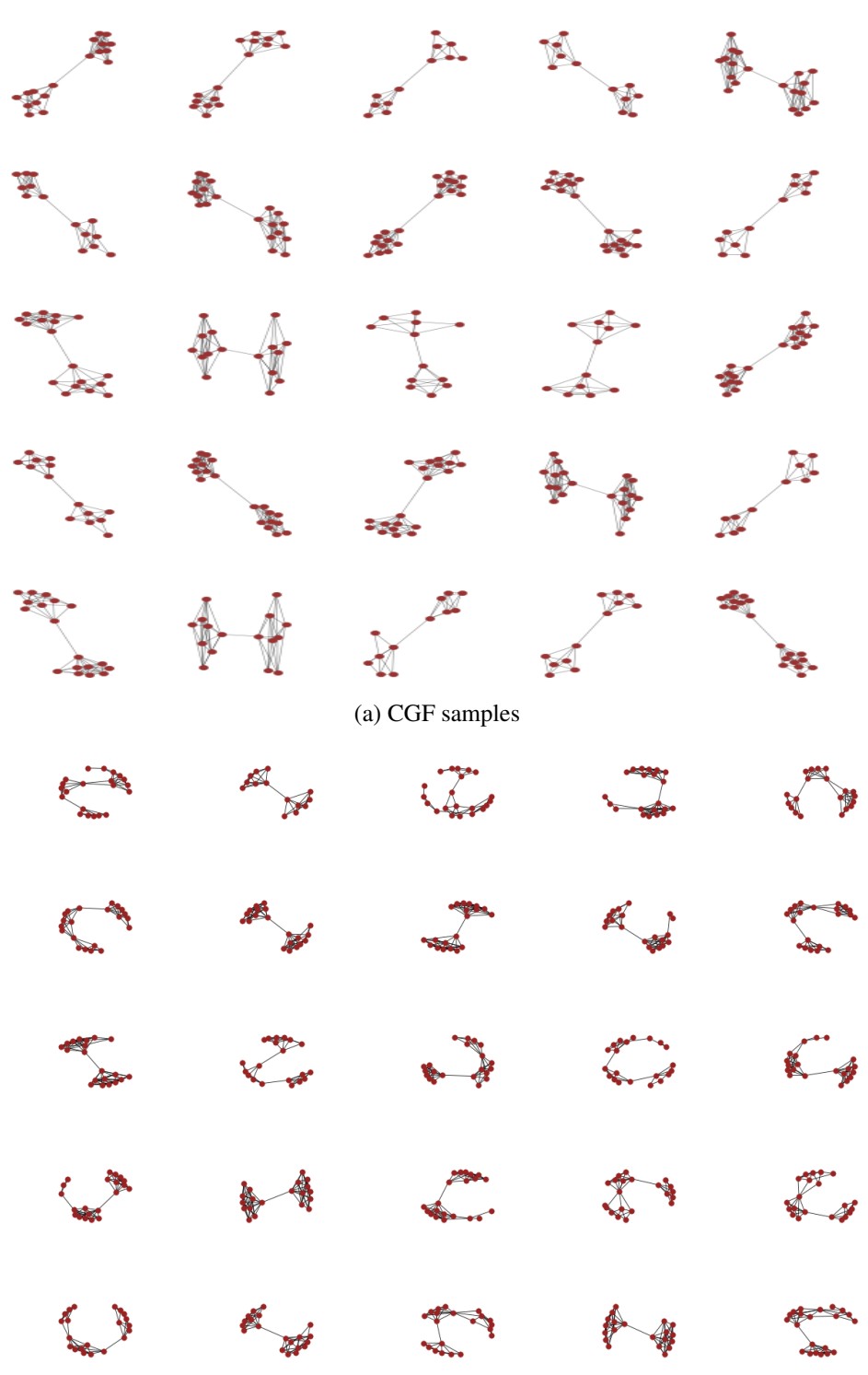

(a) CGF samples

(b) GRAPHRNN SAMPLES

Figure 12: **Graph generation on COMMUNITY-SMALL.** Samples generated using (a) our CGF and (b) GRAPHRNN.

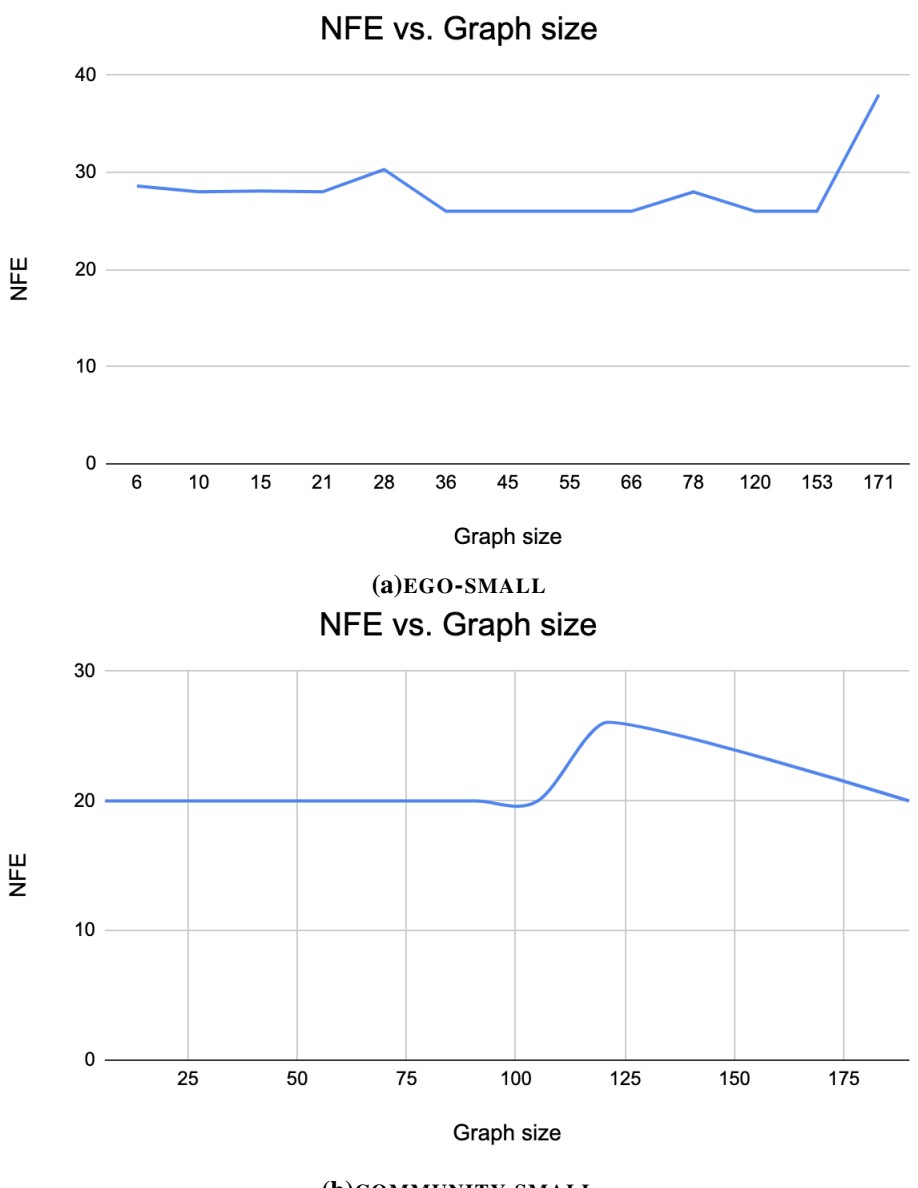

**(a)EGO-SMALL**

**(b)COMMUNITY-SMALL**

Figure 13: Visualization illustrating the effect on number of nodes in the graph on the number of function evaluations (NFE) required by the ODE solver for (a)EGO-SMALL and (b)COMMUNITY-SMALL datasets used in our graph generation model.

