# OpenReview forum: "Continuous Graph Flow"
_ICLR.cc/2020/Conference — Reject_

### Official Review · AnonReviewer1 · 2019-10-23
**Official Blind Review #1**

**Rating:** 3

**Review:**

This paper proposes continuous graph flow (CGF), a flow-based generative model for graphs.  Based on the idea to build continuous-time flows with neural ODEs, the node features are transformed into Gaussian random variables via message passing. The log-likelihood can be approximated stochastically.

I find it hard to assess the novelty of this work because 1) the algorithm looks like a trivial application of continuous-time normalizing flow to graph data using message passing algorithm, and 2) the concurrent work graph normalizing flow (GNF, Liu et al., 2019).  I failed to find any algorithmic innovation more than a mere application of continuous-time flow to graph data. The very idea of a flow-based model for graphs using the message passing algorithm may be considered as a contribution, but this is also blurred because of the concurrent work GNF.

The experiments look interesting, but there are some minor concerns. For the unsupervised graph generation task, CGF is shown to perform better than GNF, but I think the comparison here may not be fair because the results for GNF seem to be obtained from the paper uses a different way to construct initial node features to be fed into message passing. Specifically, according to the GNF paper, they inject an i.i.d. Gaussian noise as initial node features, but this work uses different schemes based on the dual graph (btw I think the term dual graph may be misleading. There already exists a common term called dual-graph with different definition). So I think to compare the expressive power of flows, it would be fair to start from a common scheme to build initial node features. GNF should also be compared to CGF for the other experiments. Seems like the baselines presented in Table 2 and Table 3 are quite dumb baselines not well suited for the tasks considered here.

In page 2, the authors stated that "GNF involves partitioning of data dimensions into two halves and employs coupling layers to couple them back. This leads to several constraints on function forms and model architectures...". I think GNF can be improved using more advanced transformation other than affine coupling, such as recently proposed mixture logistic CDF [1] or neural spline flow [2].  Continuous-time flow may still have an advantage in memory usage, but at least in terms of expressive power, I think it is not clear whether CGF is particularly better.

[1] Ho et al., Flow++: Improving Flow-Based Generative Models with Variational Dequantization and Architecture Design, 2019
[2] Durkan et al., Neural Spline Flows, 2019

**Experience Assessment:**

I have read many papers in this area.

**Review Assessment: Checking Correctness Of Derivations And Theory:**

N/A

**Review Assessment: Checking Correctness Of Experiments:**

I assessed the sensibility of the experiments.

**Review Assessment: Thoroughness In Paper Reading:**

I read the paper thoroughly.

---

> ### Author Response · Authors · 2019-11-09
> **Response to Reviewer #1**
>
> We would like to thank the reviewer for the thorough and valuable feedback for the manuscript. We updated the manuscript and the changes are marked by red color. We first list our contributions and then provide clarifications on the specific concerns listed in the reviewer’s comments.
>
> ——————————————-———-Our contributions————————————————
> We propose Continuous Graph Flow (CGF) - a continuous time flow-based model for graph structured data. In particular,
> - CGF integrates neural ODE based model and message passing (in graph neural networks) to model distributions of complex graph-structured data. Our work is among the first ones in this direction.
> - CGF learns flexible representations to handle variable data dimensions. In addition, given that CGF is based on NeuralODE, it inherits various advantages of NeuralODE such as reversibility, unconstrained functions in model design and exact likelihood computation.
>
> —————————————-——--Response to questions——————————————----
>
> *Discussion about CGF and GNF.
>
> We would like to clarify that GNF and CGF are fundamentally two different flow based models for graph-structured data. Both these models have their own advantages as well as challenges and it is valuable that these co-exist in the field.
> - Theoretical backbone: CGF is heavily entangled with Neural Ordinary Differential Equations — requires solving initial value problem and has continuous time property. GNF, on the other hand, is purely based on change of variable formula. They shall be treated differently.
> - Further improvements in models: Given that CGF and GNF rely on two different theoretical backbones, advances in the fields of Normalizing flows and Neural ODEs can potentially result in more powerful graph flow models on two independent directions. We appreciate the reviewer for pointing out the recent work on Flow++ and Neural spline flows which study improved techniques to enforce invertibility in functions. On the other hand, CGF has flexible functions in the model design. Moreover, the recent work on ordinary differential equations, such as Augmented Neural ODE [1] and Neural Jump SDE [2] can be applied to CGF or follow-up works for improving their performance, but potentially are less useful for normalizing flow-based models.
>
> *Comparison with concurrent work
> We would like to clarify that GNF is a concurrent work developed in parallel and not a previous work.  We agree with the reviewer that both these models integrate GNN and flow methods, which could be considered as key contributions of these two models. We are happy to discuss the differences between the models, but we are not claiming full contributions over GNF.
>
> *Detailed questions
> Q1. GNF uses different input data
> - The full model proposed by Liu et al(2019) consists of an autoencoder and normalizing flow with graphs as a novel and powerful architecture for handling graph generation task. We are comparing with their full model. We apologize for the confusion and we have clarified that in the table.
> - This model in the paper uses coupling layers which requires the input data to have dimension higher than one for splitting. In contrast, our model CGF can handle both one-dimensional and higher dimensional data.
> - Moreover, while the representations are not the same, the task remains the same (i.e., generating diverse graphs) and we compute the metrics on the output of the model irrespective of the model being used to represent the value of the input graph variables.
>
> Q2. Dual graph
> Thank you for pointing out. We changed the term dual graph to line graph and describe the transformation performed on the graph as edges being switched to nodes and nodes to edges. We’ve made these changes in the paper.
>
> Q3. Baselines for table 2 and table 3
> The baselines are all state-of-the-art structured and unstructured latent space baseline models — all published in the timeline 2016-2019. Our method is effective because it integrates powerful flow models for graph-structured data, and eliminate the assumptions over latent distributions (Gaussian, etc.).
>
> [1] Augmented Neural ODEs, Emilien Dupont · Arnaud Doucet · Yee Whye Teh, NeurIPS 2019
> [2] Neural Jump Stochastic Differential Equations, Junteng Jia · Austin Benson, NeurIPS 2019

---

### Official Review · AnonReviewer3 · 2019-10-23
**Official Blind Review #3**

**Rating:** 3

**Review:**

The paper presents a method to do Continuous Normalizing Flow for graphs by defining the density evolution via ODEs over shared variables. Continous message passing scheme introduced can leverage the free-form model architectures which allow flexibility in handling variable data dimensions. The results of three different generation tasks indicate improved performance over the baseline methods.

Novelty: The basic methodological contribution of the paper seems somewhat limited. The main idea of the paper follows directly from NeuralODE (Chen et al., 2018) and FFJord (Grathwohl et al., 2019). Nevertheless, I do agree that adapting the NeuralODEs and continuous flows for unstructured data is challenging and deserves its own analysis. However, I find the experiment section to be not very clear (See specific comments below).

Recommendation: Weak Reject

Concerns:

- The experiment on Graph Generation is not clear to me. What is the motivation of using the dual of the graph? What is the input to the full-connected layers? Is the input dimension N^2 (the number of edges)? Please clarify. It would be helpful if the implementation details in Appendix A.1 can be formalized in the main paper since this is the main contribution of the paper. Overall the experiment section in the main paper lacks sufficient details to comment on the efficacy of the method.

- NeuralODEs involve iterative numerical solvers which can be very slow. It seems that the proposed method involves simultaneously solving ODEs that linearly scale with the number of nodes (Equation 11)? I think there should be some analysis of how the proposed method scales with the graph size. I am not sure if NeuralODE would perform well when the number of nodes is large.

- Compared to other baseline methods (see You et al. 2018), the datasets used for analysis are smaller versions of the original datasets. There are other datasets that can be used for experiments on graph generation:  See Grid, Protein

**Experience Assessment:**

I have published one or two papers in this area.

**Review Assessment: Checking Correctness Of Derivations And Theory:**

N/A

**Review Assessment: Checking Correctness Of Experiments:**

I assessed the sensibility of the experiments.

**Review Assessment: Thoroughness In Paper Reading:**

I made a quick assessment of this paper.

---

> ### Author Response · Authors · 2019-11-09
> **Response to Reviewer #3**
>
> We would like to thank the reviewer for valuable comments and suggestions on our manuscript. We updated the manuscript and changes are marked by red color.
>
> Q1.1 Motivation of dual Graph.
> Clarification on the dual graph: We would like to clarify that the term dual graph that we previously used is confusing. We have corrected the term “dual graph” to “line graph”. This line graph is obtained by transforming the original graph with edges being switched to nodes and nodes to edges.
>
> Q1.2 The experiment on Graph Generation is not clear to me. What is the motivation of using the dual of the graph?
> CGF operates over graph nodes (variables). The task of graph generation requires modeling of edges as variables (e.g. generating the adjacency matrix). Therefore, we transform the input graph G=(V, E) to line graph G’=(V’, E’) and use CGF to directly model nodes in the line graph G’. V’ corresponds to E in the original graph G and E’ is determined by the nodes V in the original graph G.
>
> Q1.3 What is the input to the full-connected layers? Is the input dimension N^2?
> The input to the CGF is the node variables V’ from the line graph. It takes value either 0 or 1, representing non-existence or existence. The input dimension to the model is N^2 (N is number of nodes ).The fully connected layers are message functions and used for local message passing between variables.
>
> Q2. How does the proposed method scale with the graph size?
> Our CGF model scales with number of dimensions similar to FFJORD or NeuralODE. We would like to mention that although the ODE equations involving independent variables linearly scale with number of nodes. However, in the case of the graph-structured variables, they are solved collectively instead of solving them independently. Relying on graph message passing will lead to better modeling of each node and potentially lessen the computation time.
> Thank you for the suggestion. We analyzed the number of function evaluation w.r.t graph size on graph generation problem. The plots are updated in Appendix under A5. The plots show that the number of function evaluations (steps to solve ode) doesn’t linearly increase with the increment in number of nodes.
>
> Q3. Experiments on larger graphs
> Our model is generating entire graphs with a single feed-forward pass. Since our focus is on expressiveness of graph generative models, we show the potential of our model to generate graphs on small-scale graph datasets (Community-small and Ego-small). These are the established benchmark datasets for evaluating single feed-forward pass methods as done in [1,2].
> Our model could potentially be extended to larger graphs by employing autoregressive (i.e.,generating K nodes at a time) or multiscale schemes.
>
>
> [1] GraphVAE: Towards Generation of Small Graphs Using Variational Autoencoders, Martin Simonovsky, Nikos Komodakis, arXiv, 2018
> [2] Graph Normalizing Flow, Jenny Liu, Aviral Kumar, Jimmy Ba, Jamie Kiros, Kevin Swersky, arXiv, 2018

---

### Official Review · AnonReviewer2 · 2019-10-23
**Official Blind Review #2**

**Rating:** 3

**Review:**

This paper proposes a new variant of invertible flow-based model for graph structured data.
Specifically, the authors proposed a continuous normalizing flow model for graph generation, first in the graph generation literature.
The authors claim that the free-form model architecture of neural ODE formulation of continuous flow is advantageous against standard discrete flow models. Experimental results show that the proposed model is superior to recent models in image puzzling and layout generation datasets.

Overall, manuscript is well organized.
The combination of continuous flows and graph structured data is new in the literature (as far as I know). Proposed formulation seems natural and reasonable. I find no fatal flows in the formulation. In experiments, the proposed model achieved good scores against recent GNN works.

Concerning the existing invertible flow-based models for graph structured data, Madhawa’s work is one of the first attempts in the literature. I think the paper below should be refereed appropriately:
Madhawa+, “Graph NVP”, arXiv: 1905.11600, 2019.

The way of incorporating relational structure into flows are very similar to the GraphNVP and Graph Normalizing Flow: using neighboring nodes’ hidden vectors as parameters (or input to parameter inference networks).
In addition, I found no special tricks or theoretical considerations to achieve the continuous flow for graphs. Based on these points, I think technical contribution of this paper is somewhat limited.

I cannot find information about the specific chosen forms of f-hat and g in Eq.(10) within the manuscript. Are the choices of f-hat and g are crucial for performance? It is preferable if the authors can present any experimental validations concerning this issue.

My main concerns are in the experimental section.

I’m not fully convinced in the necessity of the continuous normalizing flows for the experimental tasks. None of the experimental tasks have `````'' intrinsic continuous time dynamics over graph-structured data (Sec. 2)''. Then, what is the rationale to adopt Continuous graph flow for these tasks?

One reason to adopt continuous flows is that the ODE formulation allows users to choose free-form model architecture, yielding more complicated mappings to capture delicate variable distributions.
I expect some assessments are made concerning this issue. My suggestion is (i) to test the discretized model of the proposed Continuous Graph Flow and see how the discretization deteriorate the performance, and (ii) to test several choices of f-hat and g (Eq.10) to show the necessity of ODE formulation, accommodating free-form model architecture.

In the current manuscript, Graph Normalizing Flow (GNF) is the closest competitor. However, GNF is not tested in the puzzle and the scene graph experiments. Why is that?

I’d like to hear opinions of the authors concerning these issues, and hope some discussions are included in the manuscript.


Summary
+ continuous normalizing flow is first applied to graph structure data
+ manuscript is well organized
+- natural and reasonable formulation. But at the same time, technological advancement is limited.
- Less convinced to adopt continuous flow for graph-structured data without no intrinsic continuous dynamics.
- Necessity or advantages of ``continuous’’ flows are not well assessed in the experiments. Please consider some additional assessments suggested in the review comment.
- GNF, the closest competitor, is omitted in the 2d and the 3rd experiments. No explanations about this.

**Experience Assessment:**

I have read many papers in this area.

**Review Assessment: Checking Correctness Of Derivations And Theory:**

I did not assess the derivations or theory.

**Review Assessment: Checking Correctness Of Experiments:**

I assessed the sensibility of the experiments.

**Review Assessment: Thoroughness In Paper Reading:**

I read the paper at least twice and used my best judgement in assessing the paper.

---

> ### Author Response · Authors · 2019-11-14
> **Response to Reviewer #2**
>
> We would like to thank Reviewer2 for the constructive comments regarding the experiments.
>
> *Discussion about CGF and GNF.
> GNF and CGF are fundamentally two different flow based models for graph-structured data. Both these models have their own advantages as well as challenges and it is valuable that these co-exist in the field. We highlight key differences between these models as follows.
> - Theoretical backbone: CGF is heavily entangled with Neural Ordinary Differential Equations — requires solving initial value problem and has continuous time property. GNF, on the other hand, is purely based on change of variable formula. They shall be treated differently.
> - Application: GNF and CGF can both be used for generating static graph structured data. In addition, since CGF is capable of modeling continuous time dynamics of graph-structured variables, the CGF can also be potentially applied to graph data having regular and irregular time dynamics.
>
> *Discussion about GraphNVP
> We agree that GraphNVP is a relevant related work in this field. However, we believe GraphNVP is based on normalizing flow based models which are different from CGF which is based on neural ordinary differential equations (ODEs). We have cited and discussed this work in the updated paper.
>
> *Comparison with concurrent work
> We would like to clarify that GNF is a concurrent work developed in parallel and not a previous work. We only use their numbers on graph generation experiments for better reference and don’t use it as a standard baseline. We are happy to discuss the differences between the models (in the following part), but we are not claiming full contributions over GNF.
>
>
> *Choices of f-hat and g:
> We apologize for the confusion. f_hat (x_i,x_j) is given by f_u (x_i) + f_b(x_i,x_j) for all j in the set of neighbors of node in the graph. To clarify the implementation details, we chose two types of functions for f_u and f_b as convolutional networks for puzzle graph generation and linear neural networks for scene graph based layout generation and graph generation. The function g is the mean of all the outputs f_hat used to aggregate these message functions.
>
> *CGF for continuous time dynamics for graphs (reasons):
> Thank you for the comment.
> - Continuous normalizing flow is the state-of-the-art flow-based model which yield a powerful tool for modeling graph distributions
> - The freedom on incorporating any function types as message functions. This allows the model to be general and makes it applicable to various types of graph neural networks in graph generation setting.
> - There is potential to use our model for continuous time graph data, especially on irregular timestamps. It’s a promising direction for future work.
>
> *Discrete vs. continuous experiments:
> - First of all, the functions in continuous dynamics cannot be used in discrete cases, since the discrete flow has limitation on using specific function types (reversible functions with easy to compute Jacobian determinant)
> - Secondly, for the standard discrete normalizing flow and continuous dynamic based flow, this experiments are already presented in FFJORD paper which compares the capacity of popular discrete models and the proposed continuous dynamics based flow.

---

### Public Comment · ~Anonymous_Reader2 · 2019-10-18
**Incremental contribution and a very very vague experimental section**

Since on the theoretical side there is only incremental contribution, it's necessary to have a stronger experimental results. Thus it's necessary to see the superiority of this method in a correct set of experiments. I have read through the experimental section at least 3 times and still many of the things don't make sense to me.

One important question is, why the authors compared their method with different set of baselines in each task?
I think the GraphNormalizingFlow (Liu et al., 2019) is an important baseline and should be included in all the experiments. It is not clear why this method is not included in the puzzle/layout generation tasks.

---

> ### Author Response · Authors · 2019-10-21
> **Thank you for your interest and comments**
>
> We thank the Anonymous Reader for the interest in our paper. However, we don't quite understand the comment especially on experiments since there are no specific concerns listed. We are happy to provide more information but we are not sure which parts are unclear.
>
> Regarding comparisons, to clarify, Graph Normalizing Flows by Liu et al(2019) is a concurrent work (as mentioned in Intro) rather than a standard baseline. We take the quantitative results for graph generation directly from their paper for reference only and follow the same evaluation protocol to conduct experiments for our proposed model.

---

> > ### Public Comment · ~Mr_Reviewer1 · 2019-11-07
> > **Thank you for the response -- Clarifying my concern**
> >
> > Here's the first question:
> >
> > In one set of experiments the following baselines are chosen to be compared against the proposed method
> > - GraphVAE
> > - DeepGMG
> > - GraphRNN
> > - GNF
> >
> > In another set of experiments these baselines are chosen to be compared against the proposed method:
> > - BiLSTM + VAE
> > - StructuredVAE
> > - Graphite
> > - VMP-SIN
> > - GAE
> > - NRI
> >
> > The problem is that there's no intersection between these set of baselines. Why?
> > Does this mean that the proposed method could only beat some of the baselines in each task but not all? So only those with lower performance than the proposed method are included in the tables?
> > In any case, this kind of experimental setup seems so weird to me, except there's a very particular reason for this choice of baselines.

---

### Public Comment · ~Mr_Reviewer1 · 2019-11-07
**My second question regarding the vague experimental section**

What does DEGREE, CLUSTERING, and ORBIT mean in the graph generation section?
These words (degree, clustering, and orbit) are only mentioned in Table 1 and there're no explanations of these in the text. I believe this task (graph generation) and the criteria for comparison deserve a short description, even if it is following the same experimental setup as GNF.

Why lower Orbit is better? Why lower Clustering is better? etc.

---

> ### Public Comment · ~Mr_Reviewer1 · 2019-11-08
> **Also dual graph is not the correct term to use here**
>
> In graph theory dual graph has a different meaning.
> The correct term to use here is Line graph (or edge-to-vertex dual graph) [Gross and Yellen 2006].

---

### Public Comment · ~Mr_Reviewer1 · 2019-11-07
**My third question regarding the performance of the proposed method on the puzzle generation task**

There are huge discrepancy between patches in each generated puzzle in Figure 3. It seems as if patches are generated independently of each other without passing any information among them.
I was hoping to see an ablation study on the graph structure. For example, what happens if none of the nodes (patches) in the graph are connected? Or in the opposite case, what happens if all the nodes are connected?
If the method actually works, I would expect to see more similarity between patches in the generated puzzles when there are connections between nodes and less similarity when there's no connection.
Also, what would be the NLL in those cases? Is Continuous Graph Flow actually capable of performing message passing?

Such ablation study with both qualitative and quantitative results are necessary.

Also, I wonder if a simple generative model (e.g. VAE, Flow, etc) that is trained on each patch and conditioned on the index of each patch ( top-left: 1, top-middle: 2, etc) would produce better results (NLL) than this method or not. (P(x_i | i) where x_i is a patch and i is the index of that patch. Then log-likelihood of a generated puzzle would be \Sum_{i=1}^9 P(x_i | i) ).

---

### Decision · Program_Chairs · 2019-12-19

**Decision:**

Reject

**Comment:**

Novelty of the proposed model is low. Experimental results are weak.